# Inequalities in Resource Distribution and Healthcare Service Utilization of Long-Term Care in China

**DOI:** 10.3390/ijerph20043459

**Published:** 2023-02-16

**Authors:** Changyong Yang, Jianyuan Huang, Jiahao Yu

**Affiliations:** 1Department of Sociology, Hohai University, Nanjing 211100, China; 2Population Research Institute, Hohai University, Nanjing 211100, China

**Keywords:** inequality, LTC, healthcare, resource distribution, services utilization, urban–rural

## Abstract

Background: Long-term care (LTC) services help the elderly maintain their functional ability and live with dignity. In China, the establishment of an equitable LTC system is a primary focus of the current public health reform. This paper assesses levels of equality in resources for and utilization of LTC services between urban and rural areas and economic regions in China. Methods: We use social services data from the China Civil Affairs Statistical Yearbooks. Gini coefficients against elderly population size are calculated for the number of institutions, beds, and workers, and the concentration index (CI) against per capita disposable income is calculated for the number of disabled residents per 1000 elderly people and the number of rehabilitation and nursing services per resident. Results: The Gini coefficients against the elderly population in urban areas indicate relatively good equality. In rural areas, the Gini coefficients have increased rapidly from relatively low values since 2015. The CI values in both urban and rural areas are positive, indicating that utilization is concentrated in the richer population. In rural areas, the CI values for rehabilitation and nursing have remained above 0.50 for the last three years, implying high levels of income-related inequality. The negative CI values for rehabilitation and nursing services in urban areas in the Central economic region and rural areas in the Western region imply a concentration of resource utilization toward poorer groups. The Eastern region shows relatively high internal inequality. Conclusion: Inequalities exist between urban and rural areas in the utilization of LTC services, despite similar numbers of institution and bed resources. Resource distribution and healthcare service utilization are more equal in urban areas, creating a low level of equilibrium. This urban-rural split is a source of risk for both formal and informal LTC. The Eastern region has the largest number of resources, the highest level of utilization, and the greatest internal variation. In the future, the Chinese government should enhance support for the utilization of services for the elderly with LTC needs.

## 1. Introduction

With the increasing level of population aging globally, access to long-term care (LTC) to ensure functional capacity and live with dignity is critical for older adults with declining physical functional status. Therefore, how to develop a sustainable and equitable LTC system has become one of the key policy issues of concern for many countries [1]. Such a government focus implicitly raises two concerns about the LTC system. On the one hand, people are caring about whether LTC services are a sustainable survival security system. In other words, what kind of security role does LTC play for the elderly, and will there be enough elderly people in the future who will need LTC services to intervene in their daily lives? On the other hand, people are also thinking about the policy issues regarding the equity of LTC. The current LTC system is taking up more and more public funds, but has such a model ensured or improved the equitable distribution of LTC services?

The first concern has been adequately explained by existing research. LTC is effective in compensating for functional decline in older adults and reducing the burden of home care, demonstrating its effectiveness for survival assurance [2]. According to the Organization for Economic Cooperation and Development (OECD) countries, the number of older people who will need additional assistance in their daily lives will rise sharply in the coming decades [3]. At the same time, public spending on LTC systems will increase dramatically as more countries begin to roll out long-term care financing subsidy programs (both for institutions and individuals). The EU estimates that its LTC public spending will soar from 1.4% of GDP in 2014 to 4.3% in 2060 [4]. This means that the LTC system will have a large number of clients to serve, proven effective service capacity, and stable public funding interventions. While there is no guarantee that everyone will have equal access to LTC services, it will undoubtedly facilitate the sustainability of LTC service delivery.

For the second concern, people have not found a reliable LTC system that can guarantee the fairness of LTC services at present. But for LTC, as a system with greater public resource distribution and breadth of population coverage than the traditional health care system, the availability, and accessibility of LTC’s resources must be equitable for all older adults [5]. When LTC was first known to the public, there was a view that the system left little room for inequality [6]. However, this view became increasingly unsupportable to the public as LTC was pushed out globally. People discovered that the existing LTC system had relatively wide inequalities in both resource distribution and service access and/or utilization.

In terms of resource distribution, many believe that the higher the distribution of public resources, the less inequality there will be in LTC. However, Fernandez finds that because the central government leaves too much leeway to local governments in allocating public resources and setting service standards, it breeds geographic inequality in the LTC system [7]. Rich areas are attracting more service providers by providing excessive government services and lower qualification thresholds [8]. This will squeeze the service resources in underdeveloped areas. At the same time, local governments with a social democratic orientation may be more inclined to allocate excessive public resources to the LTC system than those with a liberal capitalist orientation [9]. Recent research has also found that local governments’ differential prioritization of LTC systems in overall social security policy and differential emphasis on institutional and family services have contributed to significant differences in the geographic distribution of existing LTC systems [10,11]. These differences can hardly be explained by the changes in the population size and needs of the elderly, but the results all lead to the inequality and unfairness of LTC.

On the utilization of LTC services, the most striking finding is the socioeconomic gradient in LTC use. In most European countries, LTC services have a clear pro-rich tendency, and their use is closely related to personal savings, family income, and housing ownership. Even after controlling the health differences of the elderly, this tendency is still significant [12,13,14]. In the meantime, Rodrigues’ research found that only in informal LTC service, especially in family service, there is inequality in favor of the poor [15]. In summary, due to the extensive inequalities in the current LTC services in terms of vertical resource distribution and horizontal service utilization. Even developed countries (such as Germany, the Netherlands, and Japan), which were the first to implement LTC services, are currently reforming their LTC systems by expanding coverage and increasing public spending. However, the concrete effects of the reforms remain to be examined [16].

In addition to developed countries, the demand for LTC in many emerging countries is also growing rapidly, and the rate has even far exceeded that of developed countries. These countries are trying to establish their LTC systems. However, informal care (mainly home care) still occupies a central position in developing countries. Almost all developing countries do not have an established LTC policy and system, and there is also a lack of research on the state of LTC provision in these countries [17,18]. China, as the world’s most populous country with the fastest aging rate, is no exception. Thanks to its rapid economic development, China has made significant progress in public health [19], but better health services for the elderly are still needed. The vast majority of the health needs of elderly people are LTC needs arising from increasing physical and mental deterioration [20]. Data from China’s seventh national census shows that the number of its people aged 65 and overreached 191 million, accounting for 13.5% of the total population. Compared to its sixth census in 2010, the increase in the number of elderly people reached a staggering 60.50%. The rapid rise in the number of older adults has led to a dramatic increase in the incidence of chronic diseases and disabilities among the elderly in China [1]. Therefore, the explosive growth in demand for LTC services has become a public health service challenge that the government urgently needs to address [21]. China has been promoting the distribution of LTC resources and improving the availability of services through three consecutive national five-year plans since 2006.

At present, China has established an LTC system covering almost all the elderly at a low level. Its LTC system includes two levels: financial protection and service protection. Financial protection is the main funding and payment system for LTC, which provides all ADL-assessed elderly with the necessary funds for LTC services. It takes the form of social insurance, commercial insurance, and social relief. In terms of coverage, such a financial security system includes almost all urban elderly people. For rural elderly people, only those in financial difficulty can be supported through social relief. Social insurance is mainly paid for by residents and/or their employers and is available to all insured people with long-term disability status. In 2016, the Chinese government launched a pilot long-term care insurance system (LTCI) in 15 cities (in seven eastern cities such as Shanghai, five central cities such as Changchun, Jilin, and three western cities such as Shihezi, Xinjiang). It attempts to address the costs required for basic living care and medical care closely related to basic living for severely disabled people. Commercial insurance is paid by the residents themselves, and residents who pass the ADL assessment are entitled to a subsidy for the cost of care. Social relief is borne by the government and mainly covers elderly people in financial difficulties (with income below the average social wage and over 70 years of age). For example, in 2014, the Chinese Ministry of Finance and the Ministry of Civil Affairs together introduced the “Subsidy System for Elderly People with Disabilities in Financial Difficulties”, which aims to provide financial subsidies for these elderly people to access elderly care services. In addition, the purpose of the service protection is to guarantee LTC services in professional institutions that residents can enjoy nearby, and a subsidy system for elderly care institutions was implemented in 2014. It mainly consists of subsidies for the construction and operation of LTC institutions, as well as economic subsidies for the elderly to receive LTC services. The exact amount of which is determined by each province according to its economic development, and there is no uniform standard.

However, unlike most countries where LTC systems cover both institutions and households, LTC in China is currently only for institutions. LTC institutions provide the majority of formal LTC services and play an important role in the LTC system [22]. It is well documented that inadequate LTC increases the burden on the healthcare system [23] and that inequalities in formal LTC shift the burden of care onto families, negatively impacting older people with weak family support [14] and putting pressure on family members. Moreover, as the largest developing country, China has large regional and urban-rural disparities [24], and these are reflected in the development of its institutional LTC. The huge economic differences between the Eastern, Central, and Western regions have a substantial impact on the distribution of LTC resources [25]. The number of LTC institutions and beds available in urban areas has grown rapidly in China since the late 1990s when market players began to establish apartments for the elderly in urban areas [26]. As these institutions opened, more and more families began to accept institutionalized aging [27], and utilization of urban LTC services increased. Inequalities in economic income between urban and rural areas are extremely high [28], with the result that access to care services in rural areas is comparatively low [29].

Because of these specific issues related to LTC, an increasing number of elderly people are being excluded from the LTC system. The resulting impact not only has a significant impact on the quality of life of older adults but also affects the effectiveness and efficiency of the LTC system. It is regrettable that current research on LTC, especially in emerging countries such as China, still focuses primarily on the LTC needs of older adults and approaches to LTC system construction. However, research on the more basic and necessary LTC resource distribution and utilization inequities that precede these issues is rare. The purpose of this research is therefore to examine the evolution and equality of LTC resource distribution and service utilization in China to uncover its urban-rural and regional disparities and the socioeconomic factors that contribute to them. To our knowledge, this is the first study to assess and examine inequalities in LTC resources and service utilization at a national level, specifically introducing an urban-rural perspective in a Chinese context. Drawing on official national data, we use the Gini coefficient and the concentration index (CI) to assess inequalities in resource and service utilization. We hope that our findings will not only provide new reform ideas for China but also warn countries trying to build new LTC systems to intervene moderately in their resource distribution and utilization.

## 2. Materials and Methods

### 2.1. Data Sources

This study uses social services data from the China Civil Affairs Statistical Yearbook from 2014 to 2018, which records statistics related to social elderly service institutions in 31 provinces of China from 2013 to 2017, covering municipalities directly under central government authority and autonomous regions in mainland China. Data for Taiwan, Hong Kong, and Macao are not included in this study. The 31 provinces were divided into three regions according to their geographical distribution and GDP per capita: the Western (undeveloped) region covering Inner Mongolia, Ningxia, Gansu, Shaanxi, Sichuan, Chongqing, Guizhou, Yunnan, Guangxi, Qinghai, Xinjiang, and Tibet; the Central (developing) region covering Heilongjiang, Jilin, Shanxi, Henan, Anhui, Hubei, Jiangxi, and Hunan; and the Eastern (developed) region covering Beijing, Tianjin, Hebei, Liaoning, Shandong, Jiangsu, Zhejiang, Shanghai, Fujian, Guangdong, and Hainan [30]. Population and disposable income per capita data for the corresponding years are obtained from the China Statistical Yearbook.

Most LTC institutions are under the supervision of the Chinese civil affairs authorities. In the civil affairs statistics, institutions affiliated with elderly and disabled people that provide residential services include urban LTC institutions, rural LTC institutions, honorary homes, honorary military rehabilitation hospitals, demobilized military sanatoriums, and military rest homes. The first two are managed by the civil affairs departments, while the rest are managed by the military. Honorary homes are living care institutions exclusively for military personnel and their families. Honorary military rehabilitation hospitals, demobilized military sanatoriums, and military rest homes provide living care and rehabilitation care accommodation for disabled military personnel, demobilized military personnel, and retired cadres of the army, respectively. These four types of institutions provide resources only to military personnel (not to other personnel), and all care costs are covered by the military. According to Chinese media reports, the number of all elderly military cadres in mainland China in 2022 is about 267,000, accounting for only 0.10% of the overall elderly population, and we, therefore, do not include them in our study. Since the 2015 statistical yearbook does not provide separate data for urban and rural areas, we use the five-year average results as a proxy. Four provinces, namely, Hainan (eastern region, the elderly account for 1.13% of the Eastern region), Tibet and Inner Mongolia (western region, the elderly account for 7.14% of the Western region), and Jilin (central region, the elderly accounts for 6.65% of Central region), have serious missing data for resource utilization, and they are therefore excluded from our measurements of inequality of utilization.

### 2.2. Measurements of Inequality

The Gini coefficient and the CI are instruments commonly used to measure inequality. The Gini coefficient is a well-known general measure of inequality [31], and the CI is a measure of relative income-related inequality in health, the most commonly used inequality measure index [32]. Whereas the Gini coefficient is used to measure the distribution of LTC resources relative to the size of the elderly population, the CI is used to measure the distribution of LTC resources and service utilization relative to economic status. The Gini coefficient is a quantitative indicator of the degree of uneven distribution of wealth based on the Lorenz curve, which involves ranking a country’s total population in descending order of income, finding the percentage of the population and the corresponding cumulative percentage of resources or services, and depicting the correspondence on a graph. The Gini coefficient is thus a proportional value obtained by dividing the area of the region between the diagonal and the Lorenz curve by the entire area between the diagonal and the horizontal axis:(1)$$S=\frac{1}{2}\overset{n}{\underset{i=1}{\sum }}\left(LR_{i-1}+LR_{i}\right)\times P_{i}$$
(2)$$LR_{0}=0$$
(3)$$G=2\times \left(\frac{1}{2}-S\right)$$
where *S* is the area between the Lorenz curve and the horizontal axis, $LR_{i}$ is the cumulative percentage of LTC resources, and $P_{i}$ is the proportion of the population in each group ranked from lowest to highest in terms of LTC resources. *i* depends on the number of provinces considered. *G*, the Gini coefficient, takes a value from 0 to 1. Referring to the critical value of the Gini coefficient given by the China national bureau of statistics and the classification of the Gini coefficient threshold value in existing literature [33,34]. We divide the cut-off value of the Gini coefficient into a value of 0 denotes absolute equality in the distribution of resources; a value of 1 denotes absolute inequality; a value of *G* less than 0.3 denotes a preferred equality status; a value greater than 0.4 triggers an inequality alert; and a value exceeding 0.6 reflects a highly inequitable distribution of resources.

*CI* algebraic expressions are not unique. Drawing on the application of *CI* in health services, we adopt the formula used in a previous study [34]:(4)$$CI=\frac{2}{n\mu }cov\left(LU_{i},R_{i}\right)$$
where $R_{i}$ is the fractional rank in terms of disposable income per capita. $LU_{i}$ is the LTC utilization indicator, and μ is their mean. The value 2 is included to ensure that the *CI* ranges from −1 to +1. A value of 0 indicates absolute equality; a negative value indicates a concentration of LTC resources or services in poorer populations; and a positive value represents a concentration of LTC resources or services in richer populations [32]. *n* depends on the number of provinces considered.

### 2.3. Indicators

Long-term care is a relatively new concept in China. As for the specific service categories, according to the classification in the government report and the Statistical Yearbook, institutional LTC in China generally includes a range of services such as daily living care (assistance with daily care such as bathing and dressing), professional care (non-therapeutic care and rehabilitation services), and other services aimed at providing long-term support for people who have lost their daily living abilities and cognitive functions. In terms of specific data selection, we refer to previous literature [8,12,33,34,35,36,37] and combine the actual situation of LTC in China to establish a system of indicators to examine the distribution and utilization of LTC (see Table 1).

Most LTC is low-tech, especially in late-developing countries, where greater availability of institutions, beds, and workers plays a key role in securing LTC and quality of life [35]. Therefore, we used three indicators, the number of institutions, beds, and workers, to examine the resource distribution of LTC in China. We operationally convert these three indicators into relative indicators with comparative significance (mean value per 1000 elderly).

Previous studies on LTC utilization have generally considered whether the service has been used as an indicator of service utilization [33,36]. In China, LTC is officially defined in terms of the provision of a series of basic life care and closely related healthcare services for the disabled elderly for a period of time [37]. Considering LTC service is different from therapeutic service in the Chinese context [20], we define healthcare service utilization in institutions specifically in terms of the level of occupancy, and rehabilitation and nursing services utilized by the disabled elderly. Healthcare services utilization is measured in terms of the proportion of disabled residents per 1000 elderly people, and the number of rehabilitation and nursing services per resident.

## 3. Research Results

### 3.1. Equality in the Distribution of LTC Resources

Table 2 shows that from 2013 to 2017 the number of urban elderly people continued to increase, while per capita resources for LTC grew slowly. The number of institutions increased from 0.11 to 0.12, the number of beds increased from 15.59 to 17.54, and the number of workers increased from 1.76 to 2.12. The availability of beds and workers in the Eastern region was significantly higher than in the Central and Western regions. However, the number of institutions in the Eastern region was decreasing, and the gaps between the regions were narrowing.

In rural areas, the size of the elderly population was also growing, but the development of LTC resources was quite different from urban areas, with an obvious decline over the same period (Table 3). Specifically, the number of institutions decreased from 0.44 to 0.20, the number of beds decreased from 39.28 to 23.19, and the number of workers decreased from 2.37 to 1.54. The same trend of decreasing resources was observed in all three regions. In comparison, rural areas have more institutions and bed resources than urban areas. A rural-urban migrant is difficult to explain the urban-rural disparity of LTC, since rural migrant workers change jobs frequently [38]. A higher number of rural areas may be driven by political support of the continuous governmental resource distribution. 

The Gini coefficient for urban areas relative to the elderly population was less variable, ranging from 0.168 to 0.294 (Table 4). The Gini coefficient for the number of institutions ranged from 0.198 to 0.227; for the numbers of beds and workers, it decreased continuously from 0.282 to 0.168 and from 0.294 to 0.206, respectively, indicating greater equality and an overall trend of gradual improvement. The Gini coefficients for the distribution of LTC resources in rural areas relative to the elderly population were more variable, with all three increasing rapidly since 2015 (from 0.027, 0.119, and 0.075, respectively, to 0.184, 0.200, and 0.334), indicating an obvious increase in inequality.

The development of the Gini coefficient from 2013 to 2017 shows that LTC resource distribution in urban areas is improving, but that the distribution in rural areas gives cause for concern. The distribution of urban LTC institutions and beds is relatively equal and improving relative to the size of the aging population (all *G*-values are less than 0.3 and decreasing). Thus, urban LTC resources have basically reached a low level of equilibrium. The rate of decline in *G* values, representing rural LTC resources, has slowed. Distributional inequality has increased significantly with the end of policies that tilted resources toward rural areas. The urban-rural differences are shrinking, obviously at the expense of increasing inequality in rural areas. This implies a risk of inequality between urban and rural areas in terms of the size of the distribution of LTC resources as policy interventions wane.

### 3.2. Equality in the Utilization of LTC Resources and Healthcare Service

From 2013 to 2017, urban areas provided more LTC resources and healthcare service utilization than rural areas (Table 5), although rural areas have more bed resources. Trends in specific utilization indicators varied, with the gap between urban and rural areas widening for the number of disabled residents per 1000 elderly people, but narrowing for the number of rehabilitation and nursing services per resident.

From 2013 to 2017, the number of urban disabled residents maintained a steady increase (1.71–2.00 per 1000 elderly people), whereas the number of rural disabled residents declined and then rebounded slightly (from 0.80 to 1.16 per 1000 elderly people). Rehabilitation and nursing services per resident in urban areas increased and then decreased (ranging between 2.62 and 3.13 per capita), while rehabilitation and nursing services per resident in rural areas maintained a steady increase (1.30–1.96 per resident). However, this change is not the same as the trend in the number of workers between urban and rural areas. While the number of services is rising in both urban and rural areas, the number of workers in rural areas is declining. This may be closely related to the difference in government purchase of services policies between urban and rural areas.

LTC resources and healthcare service utilization in the Eastern region were much higher than in the Central and Western regions, and the gap widened. In urban areas, the Central region scored lower than the Western region for both LTC resources and healthcare service utilization; in rural areas, the Central economic region scored only slightly higher than the Western region. From 2016, the number of rehabilitation and nursing services per resident in the Western region surpassed that in the Central region. It is worth noting that the above changes exhibit the opposite of the regional economic level status. The central region has a higher level of economic development than the western region, but the western region has a higher utilization of services. The reason for this may be that the lower economic level makes it more common for the young population in the west to work outside their hometowns. This leaves western households with less capacity for services, forcing older adults to seek services in the marketplace.

The *CI* values for LTC resources and services in both urban and rural areas were positive, implying that LTC resources and healthcare service utilization were concentrated among the richer population in both cases. The *CI* values for disabled residents in urban areas continued to decline (from 0.483 to 0.329), implying an improvement in equality. The *CI* values for disabled residents in rural areas increased, remaining above 0.50 for the last three years of the period under study, implying a high level of income-related inequality. Such a time series with a decreasing trend of urban inequality and an increasing trend of rural inequality. There is a similar change to the trend in the Gini coefficient, implying that the improvement in LTC utilization in urban areas is built on the premise of increasing inequality in rural areas. The *CI* values for urban and rural rehabilitation and nursing services fluctuated but were low overall (ranging from 0.185 to 0.384), indicating reasonable income-related equality (Table 6). 

Within different economic zones, more complex income-related inequalities of LTC resources and healthcare service utilization were found (Figure 1). The *CI* values were positive in the Eastern region and higher in rural areas (ranging from 0.33 to 0.53). The *CI* values for rehabilitation and nursing services per resident were negative in urban areas in the Central region and rural areas in the Western region, indicating a concentration of resource utilization toward poorer groups. The *CI* values for rehabilitation and nursing services per resident in urban areas of the Central and Western regions gradually exceeded those in the Eastern region. However, for all other indicators, the Eastern region had higher (absolute) *CI* values than the Central and Western regions.

From the evolution of *CI* values, the inequality of resources and utilization exhibited in different economic zones shows a phenomenon of increasing inequality as the socioeconomic level increases. In other words, the differential allocation of resources by the government across regions has not substantially narrowed the gap in the level of services received by the elderly.

## 4. Discussion

This study examines LTC resource distribution and healthcare service utilization in China, in both urban and rural areas, and from the perspectives of the Eastern, Central, and Western economic regions. On the whole, under the influence of reform path dependence, the current promotion policy of LTC in China, although formally including the reform of social insurance and the establishment of LTCI and other measures, remains mainly in the substance of single financial support. But the adoption of these instruments by the Chinese government has led to unintended consequences. These instruments have promoted market enthusiasm for building long-term care resources, but have led to a large number of idle resources due to different policy inputs on the distribution and utilization sides. At the same time, such policy instruments have made the current LTC service utilization in China appear pro-poor phenomenon between the regions and a pro-rich phenomenon between urban and rural areas at the same time, and have not well improved the service utilization of economically disadvantaged elderly people in rural areas. This reform trap is worthy of caution for other countries attempting to build new LTC systems. 

### 4.1. Urban-Rural Differences of LTC

We find that rural areas have more institution and bed resources compared to urban areas. The higher base number of institutions and beds in rural areas cannot be explained in terms of economic level and is actually due to continuous policy support dedicated to increasing the number of beds in rural areas. During the 11th Five-Year Plan (2006–2010), the government proposed adding 2.2 million beds in rural institutions for the elderly. In the 12th Five-Year Plan period (2011–2015), the government set a requirement of 30 beds per 1000 elderly people nationwide, with 3.42 million new beds of all types, and increased policy support and resource distribution for rural areas and the Central and Western regions. The 13th Five-Year Plan period (2016–2020) again called for improvements to the supply of services for the elderly but without an emphasis on tilting resources to rural areas. Table 3 shows that the higher distribution of bed resources in rural areas reflects the implementation of these three plans. Previous studies have also observed that the degree of administrative directives and policy support, rather than of market regulation, is what mainly influences the distribution of resources, especially in rural areas [39]. However, in contrast to previous findings that more resources flow to urban areas, our results indicate a flow to rural areas [40]. Moreover, and again in contrast to earlier studies [29], we find that the greater institutional resources in rural areas lead to greater accessibility of care services in rural areas than in urban areas.

Urban areas have more resources for workers, although they have fewer bed spaces. There are no regulations on worker numbers in China in terms of policy, and the distribution of workers is influenced by labor market mechanisms, with higher wages in urban areas likely to attract more workers. Appropriate staffing is critical in the provision of care for the elderly [41], and staffing is often defined by the number of caregivers and the degree to which their caregiving skills match the needs of the elderly population. In terms of the number of caregivers, there is a large gap between China and advanced countries in East Asia with aging populations. In 2017, South Korea had nine caregivers per 1000 elderly people, while Japan had eleven [42]. In contrast to the fixed nature of beds, workers are highly mobile, and a lack of labor contract protection in China contributes to high turnover rates [43]. The higher numbers of beds and lower numbers of workers in rural areas reflect the difficulty of recruiting and retaining staff in rural LTC institutions. Staff shortages, especially of caregivers, lead to a lack of caregiver time apportioned to elderly people, which results in incomplete care [44], with a corresponding increase in adverse events such as medical errors, falls, and pressure sores [45]. In terms of competence, many existing caregivers lack caregiving skills; instead of providing geriatric care, they can meet only the basic needs of elderly people [29,46]. Thus, LTC staffing in China needs to be improved if the LTC needs of the disabled elderly population are to be met.

### 4.2. Factors Influencing LTC Utilization between Urban-Rural Areas

The level of utilization of LTC resources and healthcare services is low, with large variations between urban and rural areas with different levels of economic development. On the one hand, not only is the level of utilization of LTC resources and services in rural areas much lower than in urban areas but the rise in utilization in urban areas is accompanied by a decline in utilization in rural areas (Table 5). On the other hand, although LTC service utilization in both urban and rural China reflects a pro-rich bias, this inequality is improving in urban areas and rising in rural areas (Table 6). The reason for this phenomenon may be related to the fact that the financial security of LTC services in China is more focused on the urban elderly, resulting in ineffective cost-sharing of LTC medical services. LTC costs are particularly high [47], and there is a large difference in retirement incomes in urban and rural areas. For urban elderly people, the main payers of LTC costs are their pensions and basic healthcare insurance, while the main payers of LTC costs for rural elderly people are their families [48,49], with urban pensions being twenty times as high as rural pensions [50]. Access to pensions for the elderly in rural China has led to a significant decline in rates of co-residence with adult children [51], which is the primary means of accessing LTC for the rural elderly [52], making it difficult for them to access adequate informal care services. Unfortunately, access to pensions, despite being sufficient to have a detrimental impact on informal care, is at too low a level to support the access of rural elderly people to formal LTC. This results in multiple disadvantages for elderly people in rural areas [53] and creates difficulties in meeting both formal and informal care needs.

In addition, the LTC service protection system and LTCI system introduced by the Chinese government may have a potential role in guiding the widening gap between the rich and the poor in LTC service utilization. In Jiangsu province, for example, which is one of the leading economies in China, the LTC service protection system in Jiangsu provides a financial subsidy ranging from 300–800 RMB per month for each elderly person admitted to an LTC institution. However, this is a far cry from the average monthly cost of admission of 3600 RMB for an elderly person in Jiangsu. With this lack of targeted subsidies, the CI of LTC institutionalized seniors in urban areas decreased by 0.07 in 2015 under the service protection policy, while in rural areas it increased by 0.27. That is, urban seniors with generally good social security and rural seniors with good financial conditions are more willing to receive LTC services in institutions. China’s LTCI system, which is being piloted, also shows an imbalance between urban and rural payment capacity. For example, the LTCI pilot in Qingdao includes both urban and rural areas, with urban workers covering 10% of the cost and others covering 20% or 40% [23]. This practice violates the basic requirements of existing research on resource distribution in LTC systems [2] and does not achieve a balance between coverage and welfare. Social security arrangements in China create significant exclusions from service utilization for the poor elderly in rural areas, which is a huge systemic pitfall for LTC resources and healthcare service utilization. People with social security benefits are more likely to be able to afford LTC, and there is a huge disparity in social security benefits between urban and rural areas: 28% of urban seniors with disabilities receive generous benefits, compared to only 5% in rural areas [54]. China thus faces the problem characteristic of low- and middle-income countries of sustaining LTC policies [55], while large differences in the existing systems in urban and rural areas make it difficult to improve the rural LTC situation in the short term.

In addition, the current government subsidies in the Chinese LTC system include both the construction subsidy and operating subsidy. The construction subsidy is based on the number of beds built by the institution at the time of construction. The operating subsidy is based on the actual number of residents. We believe that the large subsidy has been used to support the expansion of facility investment by service production entities in LTC institutions, which is consistent with government practices in other service areas [56]. And yet, given the lack of supporting financial policies to encourage utilization of LTC by economically disadvantaged rural elderly, such as cost sharing of disabled elderly and improvements to service quality. Not only make the treatment of disabled elderly who need to get LTC services not treated better, but also makes them reluctant to stay in the institutions. In China, a large number of rural LTC facilities provide only basic food and housing services to control costs and lack the provision of medical care services. At the same time, some institutions cheat the operating subsidy by attracting self-care elderly (who are not covered by the LTC system) people to stay. Therefore, it’s understandable the expansion of LTC institutions based on policy requirements, particularly but not exclusively in rural areas, has generated a large number of idle beds. This ultimately led to opposite directions of development in the distribution and utilization of LTC resources in urban and rural areas, although they have similar bed resource distribution.

### 4.3. Differences in Resource Distribution and Service Utilization between Regions

Geographic disparities in healthcare resources are prevalent in most countries and contribute indirectly to the unequal spatial distribution of individual healthcare. The study of the key mechanisms responsible for this phenomenon is an important area of public health research [57]. Analysis of regional differences in resource distribution for LTC services in China has shown that LTC resources are still significantly higher in the Eastern region than in the Central and Western regions, and that bed distribution levels in the Eastern region are already similar to those in Japan and South Korea [42]. On the one hand, this is due to demographic factors such as the density of the elderly population in the Eastern region and the high number of elderly people in need of LTC. On the other hand, in the Eastern region, the economic level is higher and LTC is more market-oriented, with companies more willing to invest there. However, this inequality in resource investment is likely to further aggravate health inequality among the elderly population, perpetuating the spatial pattern of high levels in the Eastern region and low levels in the Central and Western regions in terms of overall health status [58,59].

The results of our study suggest that service utilization is affected not only by economic factors but also by macro policy regulations. We find that resource utilization is indeed higher in the Eastern region than in the Central and Western regions. The reason for this discrepancy may be related to the operating subsidy policies. Apart from the previously mentioned operating subsidy based on the number of residents, the government also provides implicit operating subsidies consisting of tax breaks and institutional water, electricity, and gas cost discounts. However, the central government has not clearly defined the target, scope, and amount of these subsidies. Cities have a relatively large degree of discretion in setting the subsidies. For example, Jiangsu province in the eastern region stipulates that LTC institutions don’t need to pay corporate income tax. At the same time, the institution is allowed to pay for the use of water, electricity, and gas at the civilian standard. In Shanxi Province, located in the central region, institutions are only exempted from 50% of corporate income tax. Meanwhile, in some western provinces, there is no water, electricity, or gas usage fee concession policy. This policy disparity allows the wealthier east to have more policy incentives, which is opposed to the inter-regional pressure to operate LTC. This ultimately leads to higher financial expenses for seniors in the central and western regions to access services, which aggravates the differences in service utilization among regions.

Furthermore, a comparison between the Central and Western regions shows that the number of services per resident in the Western region was higher than in the Central region in 2017, but that the economic situation is lower overall. This finding is inconsistent with earlier studies’ characterization of elderly care services as pro-rich [8]. Adding this finding to our examination of the relevant CI values. We can also confirm that in recent years, with the Chinese government’s policy investment in the Central and Western regions, more and more resources have been concentrated on low-income groups. China has been sharing the LTC costs of older adults in these regions through the government purchase of services to increase their healthcare service utilization and improve their health status. Such macro-level policy interventions may themselves be responsible for disparities in service utilization, as demonstrated in previous studies on health management service utilization among older adults [60].

### 4.4. Policy Implications

Based on the findings of this paper, to promote the LTC system as equitable as possible and create an age-friendly society. The government needs to contribute at both ends of resource distribution and support for utilization.

(1)Moving away from unconditional encouraging an influx of capital into the LTC system. An effective response to low bed capacity in China is to encourage institutions with occupancy rates that meet certain standards to expand. It will help to prevent the construction of institutions that meet policy needs but do not meet the needs of the elderly. At the same time, the government needs to reform the nationally planned standard for the number of beds. Allow cities to adjust the number of beds based on regional conditions flexibly. On the other hand, the type of institutional beds should be subdivided into nursing and general beds. The LTC institutions’ bed construction and the government’s LTC subsidy should focus on nursing beds. Under the premise of clear subsidy direction, the amount of subsidy for service utilization of the elderly is appropriately adjusted according to their different situations (self-care ability and financial status). In addition, the government needs to unify the implicit operating subsidies and try to establish rules for subsidy standards. In this way, the government can ensure that institutions in different regions are treated as fairly as possible in terms of the types of subsidies. On this basis, the government should clarify the requirements for institutional care staffing and encourage investment in caregivers in rural areas.(2)A mechanism for the dynamic adjustment of institutional beds and a verification mechanism for government subsidies should be established. Current subsidy policies are not well directed toward the economically disadvantaged elderly to ensure that every elderly person has equitable access to LTC services. Adjustments are needed to meet the needs of disabled elderly and to increase support for service users. By guiding the merging, transformation, and withdrawal of institutions with excessive bed idleness, and investigating the eligibility of institutionalized elderly. Not only can the pressure on LTC system operations be reduced, but also the inpatient health care costs derived from the physician-induced demand can be prevented as much as possible. Under this premise, by including the elderly with nursing care needs in the scope of government-purchased services, encouraging nursing care beds in rural nursing facilities can cooperate with community hospitals to provide rehabilitation care services to improve bed utilization and enhance service quality.

### 4.5. Limitations

This study analyzes equality in LTC resources and healthcare services utilization in China from 2013 to 2017. Due to changes in the availability of statistics, we were unable to use the most recent data. At the same time, due to the objective lack of data for the four provinces, the sample used in this study is actually based on the territory of mainland China other than these areas and may not be fully representative of the whole country. Nevertheless, developments in the level of equality of disabled residents are obvious, and the Chinese government should release more recent data so that this phenomenon can be better understood. In addition, the discussion of service utilization needs to be more detailed than has been possible in the present study; future work should refine the indicators to explore causal relationships in greater depth, for example by constructing broader indicators of institutional financial inputs or staff salary levels to analyze whether and how LTC services in China are skewed in favor of richer groups. Indicators such as urban–rural geographic space and gender could also be constructed to clarify other factors that impact resource distribution and utilization of LTC services.

## 5. Conclusions

China has a very low utilization rate of LTC resources and a large number of unused beds. There are obvious inequalities between urban and rural areas in the utilization of LTC resources and services, although institution and bed resource distributions are similar. Equality of resource distribution and utilization is better in urban areas, creating a low level of equilibrium; inequality in resource distribution and service utilization is more marked in rural areas. This ultimately leads to the phenomenon of poor rural elderly being considered in LTC resource input policies but excluded from LTC utilization policies. From a regional perspective, the Eastern region has the largest number of institutional LTC resources, the highest level of utilization, and the greatest internal variation. The Chinese government should enhance the support for the utilization of services for the elderly with LTC needs in the future.

## Figures and Tables

**Figure 1 ijerph-20-03459-f001:**
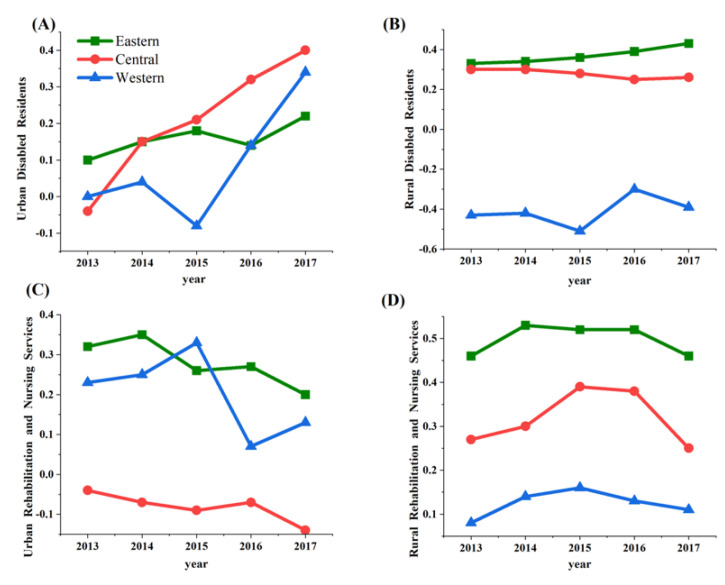
CI for LTC services in the Eastern, Central, and Western economic regions (2013–2017). (**A**,**B**) illustrate changes in CI for disabled residents in urban and rural area, respectively; (**C**,**D**) illustrate changes in CI for rehabilitation and nursing services in urban and rural area, respectively.

**Table 1 ijerph-20-03459-t001:** Evaluation index system for LTC resource distribution and utilization.

Dimension	Indicators	Definitions	Units
LTC resources	Number of institutions	Institutions per 1000 persons aged 65+	Unit
	Number of beds	Beds per 1000 persons aged 65+	Unit
	Number of workers	Workers per 1000 persons aged 65+	Persons
LTC utilization	Situation of occupancy	Disabled residents per 1000 persons aged 65+	Persons
	Rehabilitation and nursing services utilized	Rehabilitation and nursing services per resident.	Time

**Table 2 ijerph-20-03459-t002:** LTC resource distribution in urban areas (2013–2017).

Year	Economic Zone	Population Aged 65+ (Per 1000 Persons)	Institutions (Per 1000 Persons Aged 65+)	Beds (Per 1000 Persons Aged 65+)	Workers (Per 1000 Persons Aged 65+)
2013	Total	62,247	0.11	15.59	1.76
	Eastern	29,460	0.15	22.82	2.63
	Central	18,191	0.08	7.88	0.92
	Western	14,596	0.08	10.63	1.07
2014	Total	67,318	0.11	16.01	1.80
	Eastern	31,425	0.14	21.87	2.42
	Central	19,078	0.10	10.88	1.49
	Western	16,815	0.08	10.87	1.01
2015	Total	71,275	0.11	16.80	1.86
	Eastern	34,259	0.13	21.10	1.91
	Central	20,926	0.10	12.07	2.45
	Western	16,090	0.10	13.81	0.97
2016	Total	76,156	0.12	17.85	1.98
	Eastern	36,572	0.12	21.83	2.57
	Central	22,612	0.12	13.89	1.40
	Western	16,972	0.10	14.55	1.46
2017	Total	82,070	0.12	17.54	2.12
	Eastern	39,363	0.12	21.28	2.66
	Central	24,419	0.12	14.11	1.59
	Western	18,288	0.11	14.05	1.67

**Table 3 ijerph-20-03459-t003:** LTC resource distribution in rural areas (2013–2017).

Year	Economic Zone	Population Aged 65+ (Per 1000 Persons)	Institutions (Per 1000 Persons Aged 65+)	Beds (Per 1000 Persons Aged 65+)	Workers (Per 1000 Persons Aged 65+)
2013	Total	69,455	0.44	39.28	2.37
	Eastern	25,034	0.37	40.99	2.90
	Central	22,594	0.52	46.88	2.65
	Western	21,827	0.43	29.45	1.47
2014	Total	70,361	0.33	31.66	1.94
	Eastern	25,569	0.30	34.27	2.48
	Central	22,686	0.39	35.98	2.08
	Western	22,106	0.32	24.19	1.16
2015	Total	72,626	0.21	24.39	1.51
	Eastern	26,101	0.21	28.61	2.12
	Central	24,281	0.24	24.15	1.42
	Western	22,244	0.19	19.69	0.88
2016	Total	73,951	0.21	24.33	1.52
	Eastern	26,864	0.20	28.67	2.11
	Central	24,292	0.24	24.05	1.45
	Western	22,795	0.18	19.52	0.90
2017	Total	76,217	0.20	23.19	1.54
	Eastern	27,863	0.19	27.23	2.12
	Central	25,492	0.22	22.78	1.41
	Western	22,862	0.18	18.72	0.99

**Table 4 ijerph-20-03459-t004:** Gini coefficients for LTC resource distribution (2013–2017).

Year	Institutions		Beds		Workers	
	Urban	Rural	Urban	Rural	Urban	Rural
2013	0.227 [0.226, 0.231]	0.027 [0.025, 0.028]	0.282 [0.279, 0.284]	0.119 [0.117, 0.122]	0.294 [0.289, 0.300]	0.075 [0.072, 0.078]
2014	0.188 [0.186, 0.190]	0.033 [0.031, 0.035]	0.229 [0.226, 0.231]	0.134 [0.131, 0.137]	0.266 [0.264, 0.272]	0.097 [0.093, 0.101]
2015	0.196 [0.194, 0.199]	0.184 [0.181, 0.186]	0.196 [0.194, 0.199]	0.200 [0.197, 0.203]	0.237 [0.234, 0.240]	0.334 [0.330, 0.338]
2016	0.210 [0.207, 0.213]	0.190 [0.187, 0.193]	0.171 [0.170, 0.173]	0.212 [0.209, 0.215]	0.224 [0.223, 0.229]	0.341 [0.337, 0.344]
2017	0.198 [0.194, 0.202]	0.189 [0.186, 0.192]	0.168 [0.166, 0.170]	0.214 [0.211, 0.217]	0.206 [0.203, 0.208]	0.316 [0.311, 0.320]

Note: The values in parentheses are confidence intervals at a 95% confidence level for the corresponding values.

**Table 5 ijerph-20-03459-t005:** LTC resources and service utilization (2013–2017).

Year	Economic Zone	Disabled Residents (Per 1000 Persons Aged 65+)		Rehabilitation and Nursing Services (Per Resident)	
		Urban	Rural	Urban	Rural
2013	Total	1.71	1.16	2.86	1.30
	Eastern	2.79	1.45	3.20	2.45
	Central	0.64	1.26	1.64	0.77
	Western	0.69	0.73	2.51	0.70
2014	Total	1.72	0.99	2.93	1.44
	Eastern	2.75	1.36	3.52	2.60
	Central	0.76	0.97	1.19	0.87
	Western	0.71	0.56	2.52	0.78
2015	Total	1.75	0.80	3.13	1.66
	Eastern	2.66	1.24	4.19	2.90
	Central	0.86	0.68	1.10	1.04
	Western	0.85	0.39	1.52	0.77
2016	Total	1.81	0.85	2.67	1.80
	Eastern	2.76	1.31	3.16	3.02
	Central	0.83	0.71	0.89	1.05
	Western	0.93	0.42	3.17	1.11
2017	Total	2.00	0.92	2.62	1.96
	Eastern	3.01	1.45	3.18	3.67
	Central	0.88	0.67	0.79	0.88
	Western	1.16	0.50	3.04	1.07

**Table 6 ijerph-20-03459-t006:** *CIs* for LTC resources and services (2013–2017).

Year	Disabled Residents (Per 1000 Persons Aged 65+)		Rehabilitation and Nursing Services (Per Resident)	
	Urban	Rural	Urban	Rural
2013	0.483 [0.474, 0.490]	0.366 [0.354, 0.375]	0.261 [0.247, 0.272]	0.318 [0.308, 0.326]
2014	0.465 [0.455, 0.473]	0.279 [0.269, 0.287]	0.269 [0.258, 0.277]	0.185 [0.172, 0.197]
2015	0.399 [0.391, 0.405]	0.546 [0.533, 0.553]	0.384 [0.370, 0.394]	0.341 [0.326, 0.355]
2016	0.374 [0.365, 0.381]	0.599 [0.586, 0.606]	0.254 [0.245, 0.261]	0.206 [0.190, 0.222]
2017	0.329 [0.325, 0.339]	0.499 [0.487, 0.506]	0.333 [0.320, 0.377]	0.258 [0.240, 0.274]

Note: The values in parentheses are confidence intervals at a 95% confidence level for the corresponding values.

## Data Availability

China Statistical Yearbook: https://data.stats.gov.cn (accessed on 18 June 2022); China Civil Affairs Statistical Yearbook: https://www.yearbookchina.com/naviBooklist-n3022013350-1.html (accessed on 18 June 2022).

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
