# Peer review of "Inequalities in Resource Distribution and Healthcare Service Utilization of Long-Term Care in China"

_ijerph, 2023, doi:10.3390/ijerph20043459_

Round 1
Reviewer 1 Report
The study examines inequalities in the distribution of long-term care resources and the utilization of long-term care services, assessing urban, rural, and regional differences and disparities in China with the aim of generating a basis for policy-making. Therefore, a secondary data analysis is conducted. The inequality research approach is interesting, and the study is relevant from a public health perspective for securing long-term care in China.
The introduction is quite long, but very well structured and has a common thread. From the perspective of an international reader, some more nuanced information about the Chinese long-term care system and the breadth, scope, and depth of care and coverage would be desirable: what measures are in place to provide care to individuals or households in need of care? (See lines 122/123) What financing system is used? Are the costs of care services covered by the Chinese government or insurance companies, or solely out of pocket? What are the eligibility requirements and how many Chinese in need of care or elderly are covered in this case?
In describing the data sources, I would like to see a more detailed description of the exclusion of institutions and Chinese regions and the bias this may create. (lines 149-178)
The description of the main measures (G and CI) in terms of algebraic expressions seems understandable, as does the description of the interpretation of the results, which includes references. The indicators are developed in a comprehensible and justifiable manner.
The data sources, the methodological approach and the measurement concept are well described, although the data basis and the validity of the underlying data cannot be verified by me.
The results are described, interpreted and discussed in an understandable and comprehensible manner. The interpretation of the existing differences as "significant", e.g. for LTC resources (e.g. line 240), is apparently comprehensible, but methodologically unsound, since no group comparisons were made or the differences were not statistically evaluated. I would suggest replacing the term significant. In my opinion, the discussion section is too long and I recommend shortening it.
Overall, this is a well-sounded manuscript that could benefit from some minor edits and cuts.
Author Response
Dear Reviewer:
We feel great thanks for your professional review work on our manuscript entitled “Inequalities in Resource Distribution and Healthcare Service Utilization of Long-Term Care in China”(ID: ijerph-2132244). Those comments are all valuable and very helpful for revising and improving our paper, as well as the important guiding significance to our research. We have studied the comments carefully and have made corrections which we hope meet with approval. Revised portions are marked in red on the resubmitted paper. The main corrections in the paper and the responses to the reviewer’s comments are as follows:
Response to the reviewer#1’s comments:
Comment 1:
From the perspective of an international reader, some more nuanced information about the Chinese long-term care system and the breadth, scope, and depth of care and coverage would be desirable: what measures are in place to provide care to individuals or households in need of care? (See lines 122/123) What financing system is used? Are the costs of care services covered by the Chinese government or insurance companies, or solely out of pocket? What are the eligibility requirements and how many Chinese in need of care or elderly are covered in this case?
Response to comment 1:
We thank the commenters for their comments and apologize for not providing background into the policy context in which the study was conducted. We have added the appropriate policy content to this version of the manuscript. In the new text we present a new introduction to the LTC policy in China during the study period. It mainly includes he financial protection and service protection system. Currently, China has adopted social insurance, commercial insurance and social assistance to provide financial support for elderly people in need of care. Meanwhile, in terms of service protection, China subsidizes both institutions and elderly people in need of care. Construction and operation subsidies are provided to institutions with the aim of attracting them to build LTC facilities. Service subsidies are provided to the elderly in need of care with the aim of ensuring their access to a certain range of LTC services. The corresponding coverage and access conditions are also briefly mentioned in the manuscript.
The corresponding new content is added in lines 113 to 138, page3.
Comment 2:
In describing the data sources, I would like to see a more detailed description of the exclusion of institutions and Chinese regions and the bias this may create. (lines 149-178)
Response to comment 2:
We thank the reviewers for alerting us to the possible statistical bias of the article. The excluded institutions provide resources only to military personnel (not to other personnel), and all care costs are covered by the military. According to Chinese media reports, the number of all elderly military cadres in mainland China in 2022 is about 267,000, accounting for only 0.10% of the overall elderly population. The excluded provinces are Hainan (eastern region, the elderly accounts for 1.13% of Eastern region), Tibet and Inner Mongolia (western region, the elderly accounts for 7.14% of Western region), and Jilin (central region, the elderly accounts for 6.65% of Central region). However, due to the objective absence of data for these provinces, we were unable to supplement the corresponding data by interpolation. Therefore, our research is actually based on the LTC status in China outside of these four provinces. To better illustrate this point, we have added the corresponding notes in the limitations section.
The corresponding new content is added in lines 191 to 204, page 4 to 5; lines 570-572, page 14.
Comment 3:
The results are described, interpreted and discussed in an understandable and comprehensible manner. The interpretation of the existing differences as "significant", e.g. for LTC resources (e.g. line 240), is apparently comprehensible, but methodologically unsound, since no group comparisons were made or the differences were not statistically evaluated. I would suggest replacing the term significant. In my opinion, the discussion section is too long and I recommend shortening it.
Response to comment 3:
We thank the reviewer for this comment. We did have an unscientific use of language in the presentation of our conclusions. We have made corresponding changes to this content. Also, we have reduced the discussion section (especially 4.2 ) to make it closer to the results of our data results.
The corresponding new content is added in lines 425 to 470, page 11 to 12.
We tried our best to improve the manuscript and made some changes in the manuscript. And here we did not list the changes but marked them in red in the revised paper. We appreciate for reviewer’s warm work earnestly and hope that the correction will meet with approval. If there are any other modifications we could make, we would like very much to modify them and appreciate your help.
Once again, thank you very much for your comments and suggestions.
With best regards,
Authors

Reviewer 2 Report
The study aims to analyse (in)equality in the distribution of LCT resources and utilization. An interesting question. The authors use a standard methodology, but not sufficiently explain the data used (e.g., what’s the income variable used in the calculation of the CI?). The Gini indices and CIs are presented in the paper, however, the level of significance of the findings is not shown. The authors also explain the determinants of the observed urban-rural differences; however, this part of the analysis is only narrative and it is not related to the (in)equality indices. Some interesting policy implications are given, but those are not directly based on the empirical strategy followed.
Some specific comments:
The motivation of the study is too much centred on high-income countries, which are likely to have population ageing patterns and LTC systems quite different from the ones for China. Aren’t there studies done for countries with similar level of development of the LTC services and population distribution as China?
(p.3 lines 101-105) Why do you consider “elderly” people aged 60 and above rather than those aged 65 and above?
(p.4 line 176) How big are the excluded provinces (compared to the rest of the provinces in each zone)?
To what extent the urban/rural differences in LTC resource evolution can be explained by the workers mobility (rural residents moving to cities to work)?
How did you come up with the cut-off values for the Gini index (p.5 lines 197-200)?
What does the n indicator in formula (4) stand for? (p.5)
Regarding the Gini coefficients (Table 4) – are time differences within area (urban or rural) statistically significant? Moreover, Table 4 indeed reveals opposite time trends in urban and rural areas, however, it seems that the urban/rural differences are shrinking, obviously at the expense of increasing inequality in rural areas. That contradicts the statement of increasing polarizations (p.7 lines 271-272).
The same comment applies to the results for CI presented in Table 6.
Confidence intervals (and level of significance) should be provided for both Gini coefficients and CIs.
What reforms have been adopted during the study period should be explicitly mentioned and more comprehensively explained in the Introduction.
You should be very careful in generalizing your results as “exacerbated urban-rural inequalities” (p.10 lines 338-342) as that’s not what your all results reveal.
Minor
p.2 line 67 – “service access utilization” should be “service access and/or utilization”
Revise the punctuation of the whole document. Some examples of typos:
- p.3 lines 101 – “overreached” should be “over reached”
- p.5 line 199 – “distribution of resources. n depends …”
Author Response
Dear Reviewer:
We feel great thanks for your professional review work on our manuscript entitled “Inequalities in Resource Distribution and Healthcare Service Utilization of Long-Term Care in China”(ID: ijerph-2132244). Those comments are all valuable and very helpful for revising and improving our paper, as well as the important guiding significance to our research. We have studied the comments carefully and have made corrections which we hope meet with approval. Revised portions are marked in red on the resubmitted paper. The main corrections in the paper and the responses to the reviewer’s comments are as follows:
Response to the reviewer#2’s comments:
Comment 1:
The authors use a standard methodology, but not sufficiently explain the data used (e.g., what’s the income variable used in the calculation of the CI?).
Response to comment 1:
We thank the reviewer for the valuable comment. We did neglect to explain all the data sources that we used. The population and disposable income per capita data used in the CI calculations were obtained from the Chinese statistical yearbooks for the corresponding years.
The corresponding new content is added in lines 184 to 185, page 4 .
Comment 2:
The Gini indices and CIs are presented in the paper, however, the level of significance of the findings is not shown.
Response to comment 2:
We thank the reviewer for this comment. As the data we use comes from the statistical yearbook data of the Ministry of Civil Affairs of China. Its data collection method is from the census data of local civil affairs departments. Also, all institutions that provide care in China must be approved by the civil affairs department at the time of establishment and report their number of beds and caregivers. At the end of each year, they report their annual elderly occupancy status and bed utilization status. In other words, we use overall data, not sampled data, in calculating the Gini coefficients and CIs. For the full sample, the data already reflect the overall situation. There is no data error due to sampling variability, rather than the presidential situation through the sampled data. Therefore, we did not test the data for statistical significance (including confidence intervals, significance levels, etc.). However, since the purpose of our paper is to reflect the current long-term care utilization status of the elderly in China by reflecting the current status of the elderly, no group comparisons were made and no indicator system was set up for statistical evaluation based on the identification of differences. To further enhance the scientific nature of the manuscript, we normalized the wording throughout the text by replacing words such as significant.
Comment 3:
The authors also explain the determinants of the observed urban-rural differences; however, this part of the analysis is only narrative and it is not related to the (in)equality indices.
Response to comment 3:
We thank the reviewer for this comment. In the initial submission, this part of the discussion does deviate from the data results. In this revised manuscript, we have further integrated the corresponding analysis based on the actual data results.
The corresponding new content is added in lines 422 to 467, page 11 to 12.
Comment 4:
Some interesting policy implications are given, but those are not directly based on the empirical strategy followed.
Response to comment 4:
We thank the reviewer for this comment . To address this issue and to make the corresponding policy implications more coherent and follow the empirical strategy, we add a new section on Chinese LTC policies during the study period in the introduction section. The discussion section is also modified to analyze the changes in the data and the timing of policy implementation. Based on this, we have modified the policy recommendation section to follow the empirical strategy.
The corresponding new content is added in lines 113 to 138, page3; lines 555 to 558 & lines 570 to 572, page 14.
Comment 5:
The motivation of the study is too much centred on high-income countries, which are likely to have population ageing patterns and LTC systems quite different from the ones for China. Aren’t there studies done for countries with similar level of development of the LTC services and population distribution as China?
Response to comment 5:
We thank the reviewer for this comment and feel sorry for not analyzing our data in depth. To We also noticed this problem in the process of research. Regrettably, there are few peer-reviewed papers for comparison because informal care such as home care is still predominant in low- and middle-income countries, and there are no established LTC policy and system. In recent years, some papers have pointed this out, and one of the goals of this research is to supplement the research on the current situation of LTC in low- and middle-income countries. We also added this point in the revision of the manuscript.
The corresponding new content is added in lines 96 to 99, page 2.
Some references:
Peter Lloyd-Sherlock, Beyond Neglect: Long-term Care Research in Low and Middle Income Countries, International Journal of Gerontology, 2014, Volume 8, Issue 2, https://doi.org/10.1016/j.ijge.2013.05.005.
Glinskaya, E., Feng, Z., and Suarez, G. Understanding the “state of play” of long-term care provision in low- and middle-income countries. International Social Security Review, 2022, 75: 71– 101. https://doi.org/10.1111/issr.12308.
Comment 6:
(p.3 lines 101-105) Why do you consider “elderly” people aged 60 and above rather than those aged 65 and above?
Response to comment 6:
We thank the reviewer for this comment. Since some official Chinese documents use the age of 60 to define the elderly population, we have habitually substituted this concept in our writing. We have changed it to the 65 years of age discriminant for the convenience of international readers.
The corresponding new content is modified in lines 104 to 107, page 3.
Comment 7:
(p.4 line 176) How big are the excluded provinces (compared to the rest of the provinces in each zone)?
Response to comment 7:
We thank the reviewer for alerting us to the possible statistical bias of the article. The excluded provinces are Hainan (eastern region, the elderly accounts for 1.13% of Eastern region), Tibet and Inner Mongolia (western region, the elderly accounts for 7.14% of Western region), and Jilin (central region, the elderly accounts for 6.65% of Central region). However, due to the objective absence of data for these provinces, we were unable to supplement the corresponding data by interpolation. Therefore, our research is actually based on the LTC status in China outside of these four provinces. To better illustrate this point, we have added the corresponding notes in the limitations section.
The corresponding new content is added in lines 202 to 204, page 4 to 5; lines 570-572, page 14.
Comment 8:
To what extent the urban/rural differences in LTC resource evolution can be explained by the workers mobility (rural residents moving to cities to work)?
Response to comment 8:
We thank the reviewer for this comment. In fact, for China, LTC service workers are composed of two types of personnel. One is the self-recruited caregivers from private enterprises, who are usually female laborers over 40 years old with low education and professionalization, taking care of the elderly in a non-professional way and daily life. These people are often responsible for the daily care within their own families and are not able to achieve higher returns through urban-rural migration due to the human capital return mechanism (Zhang, 2011).Another part of the LTC workforce in China consists of health care professionals, and are practitioners who belong to the government public service. They are responsible for the professional care of the elderly. These personnel are not only regulated by the local government, they are a type of government employee in terms of status, and when they enter the institution through the state examination, it is basically impossible for them to move between regions except for resigning from their positions. At the same time, influenced by the difference in employment inside and outside the state system under China's redistribution system, these people also rarely return to reach inter-regional mobility by resigning. Therefore, it is difficult to use labor mobility to explain the difference in LTC resources between urban and rural areas.
The corresponding new content is added in lines 284 to 285, page 7.
Reference:
Zhang, C, Why are the rural migrant workers so prone to job change: job mobility of rural migrant workers within the constraint of hukou system. Chin. J. Sociol. 2011,6, 153–177. doi: 10.15992/j.cnki.31-1123/c.2011.06.007. (in Chinese)
Comment 9:
How did you come up with the cut-off values for the Gini index (p.5 lines 197-200)?
Response to comment 9:
We thank the reviewer for this comment. For the determination of the cut-off values of the Gini index, we referred to both the cut-off values of the Gini coefficient given by the United Nations Development Programme (UNDP) and the classification of the cut-off values of the Gini coefficient in the previous literature on health and care inequality studies. On this basis, the cut-off values of Gini coefficient in this study are determined.
The corresponding new content is added in lines 225 to 227, page 5.
Reference:
Theodorakis PN, Mantzavinis GD. measuring health inequalities in Albania: a focus on the distribution of general practitioners. Hum Resour Health. 2006;4:5.
Comment 10:
What does the n indicator in formula (4) stand for? (p.5)
Response to comment 10:
We thank the reviewer for this comment. The n indicator in formula (4) depends on the number of provinces considered.
The corresponding new content is added in lines 238 to 239, page 5.
Comment 11:
Regarding the Gini coefficients (Table 4) are time differences within area (urban or rural) statistically significant? Moreover, Table 4 indeed reveals opposite time trends in urban and rural areas, however, it seems that the urban/rural differences are shrinking, obviously at the expense of increasing inequality in rural areas. That contradicts the statement of increasing polarizations (p.7 lines 271-272).
Response to comment 11:
We thank the reviewer for this comment. As in our response to COMMENT 2, Since our data is a full sample data, it reflects the overall situation already. That is, the trend shown by the Gini coefficient is the real situation in this period. Therefore, we did not perform a statistical test on it.
At the same time, we did neglect to address the issue about the difference between urban and rural areas of variation and made a mistake in our presentation. With your reminder, we have corrected it.
The corresponding new content is added in lines 305 to 308, page 8; lines 345 to 348,page 9.
Comment 12:
Confidence intervals (and level of significance) should be provided for both Gini coefficients and CIs.
Response to comment 12:
We thank the reviewer for this comment. The response to this question is the same as comment 2.
Comment 13:
What reforms have been adopted during the study period should be explicitly mentioned and more comprehensively explained in the Introduction.
Response to comment 13:
We thank the reviewer for their comments and apologize for not providing background into the policy context in which the study was conducted. We have added the appropriate policy content to this version of the manuscript.
The corresponding new content is added in lines 113 to 138, page3.
Comment 14:
You should be very careful in generalizing your results as “exacerbated urban-rural inequalities” (p.10 lines 338-342) as that’s not what your all results reveal.
Response to comment 14:
We thank the reviewer for this comment. We did have an unscientific use of language in the presentation of our conclusions. We have made corresponding changes to this content.
The corresponding new content is added in lines 372 to 384, page 10 to 11.
Some minor comments:
p.2 line 67 – “service access utilization” should be “service access and/or utilization”
Revise the punctuation of the whole document. Some examples of typos:
p.3 lines 101 – “overreached” should be “over reached”
p.5 line 199 – “distribution of resources. n depends …”
Response to minor comments:
We thank the reviewer for this comment. We have checked the possible mistakes in the manuscript and made some modifications.
The corresponding new content is modified in line 67, page 2; line 105, page 3; linen 223, page 5.
We tried our best to improve the manuscript and made some changes in the manuscript. And here we did not list the changes but marked them in red in the revised paper. We appreciate for reviewer’s warm work earnestly and hope that the correction will meet with approval. If there are any other modifications we could make, we would like very much to modify them and appreciate your help.
Once again, thank you very much for your comments and suggestions.
With best regards,
Authors

Round 2
Reviewer 2 Report
Thanks for your reply. You’ve made a significant effort to address most of my comments.
I just outline a couple that I think that you need to reconsider (comment 9 and 2).
The first one is related to the critical values of the Gini coefficient (page 5). You mention the UNDP as a source of the classification, but not specific reference is given. And the paper you cite (ref.34) is not very relevant. The Gini index in this paper is below 0.3 and the authors talk about unequal distribution. While according to your critical values, it would mean “preferred equality status”.
And the second one has to do my previous comment 2. There is no consensus of whether the level of significance should be reported when using census data. However, most peer-reviewed papers do report (at least) confidence interval for Gini, CIs, Atkinson, Theil, etc. So, I would insist on providing this information for statistical soundness.
Author Response
Dear Reviewer:
We feel great thanks for your professional review work on our manuscript entitled “Inequalities in Resource Distribution and Healthcare Service Utilization of Long-Term Care in China”(ID: ijerph- 2132244). Based on your latest comments, we have changed our manuscript again. Revised portions are marked in red on the resubmitted paper. The main corrections in the paper and the responses to the reviewer’s comments are as follows:
Response to the reviewer#2’s comments:
Comment 1:
The first one is related to the critical values of the Gini coefficient (page 5). You mention the UNDP as a source of the classification, but not specific reference is given. And the paper you cite (ref.34) is not very relevant. The Gini index in this paper is below 0.3 and the authors talk about unequal distribution. While according to your critical values, it would mean “preferred equality status”.
Response to comment 1:
We thank the reviewer for the valuable comment. We first consulted the UNDP document on the range of the Gini coefficient, but did not find the corresponding exposition in its official documents, only in Wikipedia, where the UNDP published this coefficient range. To ensure scientific accuracy, we removed this sentence. Also, we changed the corresponding documents (ref 34 & 35). Ref.34 is the interpretation of the Gini index by the Chinese National Bureau of Statistics in 2021, while ref.35 is a peer-reviewed paper that uses the same Gini index.
The corresponding new content is added in lines 226 to 228, page 5.
References:
1.China National Bureau of Statistics. how to understand the Gini coefficient, http://www.stats.gov.cn/ztjc/zthd/lhfw/2021/rdwt/202102/t20210225_1813990.html. (Accessed Feb 12,2023)
2.Zhang T, Xu Y, Ren JP, Sun LQ, Liu CJ. Inequality in the distribution of health resources and health services in China: hospitals versus primary care institutions. Int J Equity Health (2017) 16:42. doi: 10.1186/s12939-017-0543-9
Comment 2:
And the second one has to do my previous comment 2. There is no consensus of whether the level of significance should be reported when using census data. However, most peer-reviewed papers do report (at least) confidence interval for Gini, CIs, Atkinson, Theil, etc. So, I would insist on providing this information for statistical soundness.
Response to comment 2:
We thank the reviewer for this comment. We calculated confidence intervals for the corresponding values at the 95% confidence level based on the Jackknife method proposed by Quenouille (1949). We calculated confidence intervals for the corresponding values at the 95% confidence level based on the Jackknife method proposed by Quenouille (1949). At the same time, we changed the original two decimal places to three decimal places in order to ensure that the calculated values would not cause misunderstanding among readers.
The corresponding new content is added in lines 299 to 301, page 8 (table 4),lines 354 to 356, page 10 (table 6)
References:
1.Quenouille, M. H. (1949). Approximate tests of correlation in time series. J. Royal Statis. Soc., Series B 11, 68-84.
Lian H.Q., Gao Q.H. & Zhou Y. What was the effect of personal income tax cuts on the gap between rich and poor? —Dynamic evaluation based on CHNS survey data and Jackknife method, China Joural of Economics, 2018 5(3):142-168(in Chinese)
We tried our best to improve the manuscript and made some changes in the manuscript. We appreciate for reviewer’s professional work earnestly and hope that the correction will meet with approval. If there are any other modifications we could make, we would like very much to modify them and appreciate your help.
Once again, thank you very much for your comments and suggestions.
With best regards,
Authors
